# Epicardial Adipose Tissue: Clinical Biomarker of Cardio-Metabolic Risk

**DOI:** 10.3390/ijms20235989

**Published:** 2019-11-28

**Authors:** Alexandra C. Villasante Fricke, Gianluca Iacobellis

**Affiliations:** 1Division of Endocrinology, Diabetes and Metabolism, Department of Medicine, University of Miami, Miller School of Medicine, Miami, FL 33136, USA; 2Clinical Medicine Division of Diabetes, Endocrinology and Metabolism University of Miami, Miller School of Medicine, Miami, FL, USA 1400 NW 10th Ave, Dominion Tower Suite 805-807 Miami, FL 33136, USA

**Keywords:** epicardial fat, epicardial adipose tissue, cardiometabolic risk, atherosclerotic cardiovascular risk, metabolic syndrome, visceral adipose tissue, psoriasis, HIV, race

## Abstract

Epicardial adipose tissue (EAT) is part of the visceral adipose tissue (VAT) that surrounds the heart and it is a quantifiable, modifiable, and multifaceted tissue that has both local and systemic effects. When EAT is enlarged, EAT contributes to atherosclerotic cardiovascular disease (ASCVD) risk and plays a role in the development of metabolic syndrome (MetS). In this review, we will discuss the role of EAT in various facets of MetS, including type 2 diabetes mellitus (T2DM) and insulin resistance. We examine the association between EAT and liver steatosis. We also address the correlations of EAT with HIV therapy and with psoriasis. We discuss racial differences in baseline EAT thickness. We conclude that EAT measurement serves as a powerful potential diagnostic tool in assessing cardiovascular and metabolic risk. Measurement of EAT is made less costly, more convenient, and yet accurate and reliable by transthoracic echocardiography. Furthermore, modification of EAT thickness has therapeutic implications for ASCVD, T2DM, and MetS.

## 1. Epicardial Adipose Tissue

Epicardial adipose tissue (EAT), also referred to as epicardial fat, is the true visceral adipose tissue (VAT) of the heart. EAT is located between the myocardium and the visceral pericardium and it is commonly found in the atrioventricular and interventricular grooves of the adult human heart [1,2,3,4,5,6,7,8,9]. EAT is anatomically and functionally contiguous with the myocardium (referred to as myocardial EAT) and the coronary arteries (referred to as pericoronary EAT) and it is supplied by the coronary arteries [1,2,3,4,5,6,7,8,9]. EAT originates from splanchnopleuric mesoderm [1,2,3,4,5]. EAT is made up of adipocytes, ganglia, nerves, and inflammatory, stromovascular, and immune cells [1,9]. Epicardial adipocytes are smaller than adipocytes in subcutaneous and other visceral adipose depots owing to the larger number of pre-adipocytes compared with mature adipocytes [1,10].

Physiologically, EAT serves a cardioprotective role by providing mechanical protection, serving as an energy source to the myocardium, and producing anti-inflammatory adipokines [1]. It is also hypothesized to function like brown adipose tissue and generate heat in response to cold temperatures and activate the autonomic nervous system [1,11]. EAT makes up 20% of the heart mass under physiologic conditions. EAT can, however, be associated with pathology, particularly when enlarged. Pathologic mechanisms of EAT are thought to be mediated by vasocrine or paracrine secretion of proinflammatory adipokines and free fatty acids which are facilitated by the lack of fascia separating the EAT and the myocardium [1]. Increased EAT mass is known to cause local cardiac pathology as well as systemic effects, increasing the risk of metabolic syndrome (MetS), as discussed further below.

EAT is a highly metabolically active tissue [1,2,3,4,5]. EAT synthesizes, produces and secretes bio-active molecules which are then transported into the adjacent myocardium through vasocrine and/or paracrine pathways. Given the lack of anatomical barriers separating the two tissues, the EAT secretome goes directly into myocardium and coronary lumen. Based on this evidence, EAT can be correctly considered an endocrine organ with local effects on the heart.

Two-dimensional EAT thickness can be measured by echocardiography, an inexpensive, noninvasive, readily available yet accurate and reproducible technique [12,13]. Measuring EAT thickness can be a useful diagnostic tool. Furthermore, EAT is modifiable and can be a therapeutic target. EAT can be a proxy for determination of overall visceral adiposity. It is known to be a marker of cardiovascular risk and a risk factor for metabolic syndrome [13].

## 2. Metabolic Syndrome

Metabolic syndrome (MetS), also referred to as insulin resistance syndrome, is a constellation of metabolic derangements that increase cardiovascular risk. The first definition was proposed in 1998 by the World Health Organization (WHO) [14], followed by a definition in 2001 by the National Cholesterol Education Program (NCEP) [15]. Other definitions were proposed until the most recent worldwide harmonizing criteria were made in 2009 as a revision to the NCEP criteria in conjunction with the International Diabetes Federation, the National Heart, Lung, and Blood Institute, the American Heart Association, the World Heart Federation, the International Atherosclerosis Society, and the International Association for the Study of Obesity [16]. Under the 2009 definition, the five established components of MetS include: abdominal obesity defined by ethnicity-specific waist circumference, elevated blood pressure, impaired fasting glucose, increased triglyceride levels, and decreased high-density lipoprotein (HDL) cholesterol levels. There is some controversy; for example, the American Diabetes Association and the European Association for the Study of Diabetes have stated that the existing criteria for MetS do not meet the definition of a syndrome [17]. In 2010, the WHO indicated that the criteria for MetS were an educational concept for a premorbid condition rather than a diagnostic or therapeutic tool [18].

## 3. Adipose Inflammation in Metabolic Syndrome

The development of MetS is inherently linked to the metabolic activity of adipose tissue; namely via adipose tissue inflammation, aberrant accumulation of ectopic lipid deposits, and mitochondrial dysfunction, particularly in skeletal muscle and liver [19]. MetS is proposed to be a systemic manifestation of adipose tissue disease [19]. Adipose tissue secretes many factors including leptin and adiponectin as well as proinflammatory tumor-necrosis factor-alpha and interleukins. Inflammation leads to changes in VAT including activated lipolysis, release of free fatty acids, hypoxia, oxidative stress, and apoptosis of adipocytes [20]. Specifically, increased secretion of TNF-alpha leads to increased aggregation of activated macrophages from bone marrow and infiltration of macrophages into adipose tissue [21,22]. Cell oxidative stress within the mitochondria, nucleus, and endoplasmic reticulum due to excess delivery of fuel lead to mitochondrial dysfunction [19,23,24]. Additionally, there is some evidence to suggest that opioids can trigger insulin resistance is people with a familial predisposition to obesity via the beta-endorphin system [25].

## 4. EAT as Marker of Visceral Adiposity

Visceral obesity or increased visceral adipose tissue (VAT), also referred to as visceral fat, is the adipose tissue that surrounds organs, as opposed to subcutaneous adipose tissue. Visceral obesity is associated with metabolic disorders including insulin resistance, impaired glucose tolerance, T2DM, and polycystic ovarian syndrome [26]. Increased VAT is known to play a key role in the development of metabolic syndrome [27,28,29,30,31,32,33,34]. It is also associated with cardiovascular disease including hypertension, heart failure, coronary artery disease (CAD), valvular disease, and arrhythmias; pulmonary disease including sleep apnea and emphysema; brain disease including stroke and dementia; various cancers; and reduced bone density [26]. Furthermore, VAT is an independent predictor of mortality in males [35].

Anthropomorphic measurements used to estimate VAT are imprecise, with waist circumference being the best anthropomorphic predictor of intra-abdominal fat mass [36,37]. Imaging techniques are more precise for measuring VAT, with magnetic resonance imaging (MRI) being the gold standard [38,39,40,41,42]. However, MRI is costly, time-intensive, and difficult for patients with claustrophobia to undergo. Computed tomography (CT) imaging is also another method of measuring VAT [43,44], but it is costly and required exposure to radiation. Abdominal ultrasound has been used to measure abdominal VAT but can be confounded by subcutaneous adipose tissue in obese subjects [45,46,47]. Iacobellis et al. developed a method of measuring EAT thickness via transthoracic echocardiographic ultrasound as a proxy for determination of overall VAT [12,48]. They found that EAT independently and accurately correlates with intra-abdominal VAT as measured by MRI and does so better than waist circumference does [12,48].

In the technique described by Iacobellis et al., EAT thickness was measured on the free wall of right ventricle from both parasternal long-axis and short-axis views [12,48]. Measurement of EAT on the right ventricle was chosen because this point is recognized as the highest absolute epicardial fat layer thickness [49], and parasternal long- and short-axis views allow the most accurate measurement of EAT on the right ventricle with optimal cursor beam orientation in each view. Even when hypertrophy of the right ventricular was noted, this did not confound EAT measurement [49].

The ultrasound assessment of the EAT thickness has several intuitive advantages, but also some limitations. It provides a linear measurement rather than a volumetric one. The ultrasound is operator-dependent, which could influence the reproducibility. Furthermore, EAT located in the atrioventricular groove or other locations cannot be reached with the ultrasound. Severely obese patients with MetS may have a poor acoustic window not allowing an optimal visualization of the EAT thickness. Multidetector CT and Cardiac MRI imaging can certainly provide a more accurate and volumetric measurement of EAT. EAT attenuation as measured by CT has also recently emerged as a marker of adipose tissue inflammation and a potential predictor of CAD. However, these procedures are more invasive and costly, and may not be readily available for a rapid and practical VAT assessment in the MetS patients.

## 5. EAT and Metabolic Syndrome

VAT is known to be associated with MetS. In 60 healthy subjects without diabetes, hypertension, dyslipidemia or metabolic disease, those with predominant VAT accumulation shower higher EAT thickness than those with predominant peripheral fat distribution [48]. In 72 subjects, some of which had metabolic disorders, those with predominant central VAT accumulation and at least two clinical and metabolic parameters of MetS showed higher EAT thickness on echocardiography than subjects with predominant peripheral fat distribution and no clinical or metabolic alterations [12]. EAT thickness was found to have a positive linear correlation with diastolic blood pressure, fasting insulin, low-density lipoprotein (LDL) cholesterol, glucose, and systolic blood pressure [12]. EAT was found to have a negative linear correlation with plasma adiponectin and HDL cholesterol [12]. Diastolic blood pressure and fasting insulin levels were the variables most strongly correlated with EAT thickness, as per multiple regression analysis [12]. No correlation was found with EAT and plasma triglycerides, c-reactive protein, fibrinogen, heart rate, uric acid, or microalbuminuria in this study [12].

In a 2012 meta-analysis of nine studies on echocardiographic EAT thickness in patients with and without MetS pooling 2027 subjects from nine studies, of whom 1030 had MetS, EAT thickness was significantly higher in those with MetS [50]. This meta-analysis found that patients with MetS tended to be older and that MetS was distributed evenly among men and women. It also found that the difference in EAT thickness was more pronounced in non-Hispanic White subjects followed by Hispanic, Turkish, and Asian subjects [50]. In one included study of 246 subjects, 58% of which had MetS, EAT was significantly thicker in both men and women with MetS compared to those without MetS [51]. These studies suggest that measurement of EAT, and by proxy VAT, represents an effective approach to identify patients at high risk for MetS and its subsequent consequences (Figure 1).

## 6. EAT and Cardiovascular Risk

EAT has been associated with the risk of atherosclerotic cardiovascular disease (ASCVD) including CAD [20,26,52]. Cardiovascular disease remains the highest cause of mortality worldwide according to the WHO. EAT thickness is also correlated with the development of atrial myopathy leading to atrial fibrillation and thromboembolic stroke; this is well established [53,54,55]. Early reports associate EAT with bi-ventricular hypertrophy, and impaired bi-ventricular diastolic relaxation and filling leading to heart failure with preserved ejection fraction [1,56], but more recent data is less suggestive of the role of EAT in heart failure [55]. EAT volume is not only associated with CAD but with vulnerable plaque components, which may contribute to acute coronary syndrome [20,57,58]. EAT size was also found to be correlated with left ventricular mass [59].

Crosstalk between insulin and angiotensin signaling systems may be the link between obesity and hypertension [60,61,62,63]. In a study of 374 subjects with angina pectoris, VAT as measured by CT was associated with hypertension, dyslipidemia, MetS, and coronary atherosclerosis while increased EAT was associated with low serum adiponectin, dyslipidemia, and MetS [64]. In a study comparing 157 non-obese patients with primary hypertension and 101 controls, EAT volume was greater among hypertensive patients [65].

## 7. EAT, T2DM, and Insulin Resistance

Increased VAT is associated with the development of type 2 diabetes mellitus (T2DM). Adipose expression of TNF-alpha is directly linked with the development of insulin resistance in obesity [66]. EAT is increased in people with T2DM independently of BMI or total body fat [67]. EAT has a greater capacity for uptake and release of free fatty acids compared to other visceral adipose depots and it also has a lower rate of glucose utilization [1,17]. Animal studies have shown higher rates of insulin-induced lipogenesis in EAT compared to other visceral adipose depots [1,17].

In a study of thirty obese subjects without a history of metabolic, cardiovascular, pulmonary, or hepatic disease, EAT thickness was associated with insulin resistance and impaired glucose tolerance [68]. In this study, all subjects underwent transthoracic echocardiography, a euglycemic hyperinsulinemic clamp to estimate insulin sensitivity, and an oral glucose tolerance test to evaluate glucose tolerance [68]. Among these subjects, three were found to have impaired fasting glucose, seven had impaired glucose tolerance, and twenty had no evidence of dysglycemia [68]. In particular, EAT was significantly correlated with waist circumference, fasting insulin, BMI, 120 min insulin, fasting glucose, and area under the curve for insulin even after adjusting for BMI and waist circumference in this study. No correlation was found between EAT thickness and age, area under the curve for glucose, area under the curve for insulin to glucose ratio, triglyceride to HDL cholesterol ratio, and 120 min glucose levels in this study [68].

## 8. EAT and Liver Steatosis

Hepatic steatosis or liver steatosis, also known as fatty liver disease, is the accumulation of adipose tissue in and around the liver. Hepatic steatosis, a type of VAT, provides a significant correlation with all components of MetS, independent of BMI [69]. Elevations in alanine aminotransferase [70] and gamma glutamyltransferase [71] have been found to predict MetS. Hepatic steatosis is also known to be associated with increased cardiovascular risk including CAD [72], increased coronary artery calcium score [73], atrial fibrillation [74], and overall cardiovascular risk in diabetics [75,76].

In 2014, Iacobellis et al. compared 62 obese subjects with ultrasonographic evidence of non-alcoholic fatty liver disease (NAFLD) with 62 obese controls and found for the first time that EAT was significantly thicker in subjects with non-alcoholic fatty liver disease [77]. In that study, among waist circumference and BMI, EAT thickness correlated most closely with liver steatosis. At that time, it had been known that EAT was associated with increased alanine transaminase levels [78] and that EAT and hepatic steatosis were correlated with abdominal adiposity and hypertriglyceridemia [79]. It was also known that patients with fatty liver had abnormal left ventricle energy metabolism [80] and reduced coronary flow reserve [81]. More recently, Turan found that non-alcoholic fatty liver disease fibrosis score is related to EAT thickness and CAD [82]. A recent meta-analysis of thirteen case control studies looking at 2260 subjects found that EAT was significantly increased in subjects with non-alcoholic fatty liver disease compared to controls [83]. The increase in the EAT was associated with the severity of hepatic steatosis, hepatic fibrosis and cardiovascular disease in patients with non-alcoholic fatty liver disease [83].

## 9. EAT and HIV

CAD is a known cause of premature death in people with HIV. Furthermore, antiretroviral therapy is known to be associated with MetS and lipodystrophy. EAT has been found to be increased in people infected with HIV on highly active antiretroviral therapy [84,85]. EAT thickness in patients with HIV was correlated with waist circumference, HDL cholesterol level, and plasma triglyceride level [84]. EAT thickness may be related to duration of anti-retroviral therapy as well as markers of chronic inflammation [86].

In a 2014 study of 579 HIV-infected and 353 HIV-uninfected men aged 40–70 using CT to measure EAT and coronary artery calcium, HIV-infected men were found to have greater EAT than uninfected men [87]. EAT volume was associated with duration of antiretroviral therapy, specifically azidothymidine. EAT was associated with the presence of coronary artery plaque and noncalcified plaque. In men with positive coronary artery calcium, EAT was associated with the extent of coronary artery calcium [87]. In a 2012 study of 583 HIV-infected men, EAT and VAT, but not body mass index, were associated with cardiovascular disease [88]. In a cross-sectional study of 876 HIV-infected men and women on antiretroviral therapy, EAT as measured by CT was associated with central fat accumulation, mixed lipodystrophy phenotype, cumulative exposure to antiretroviral therapy, and coronary artery calcium among other factors [89]. In a study of 78 HIV-infected men and 32 HIV-negative controls, EAT was associated with fasting glucose and plasma insulin levels [90].

The first study looking at EAT in the HIV-infected populated was a 2007 study by Iacobellis et al. in which 60 HIV-infected subjects on antiretroviral therapy, MetS and lipodystrophy were compared with 45 HIV-infected subjects [91]. Their EAT and carotid intima–media thickness were measured by ultrasonography and MRI was used to calculate VAT [91]. EAT by ultrasound was found to correlated with VAT by MRI. Patients with HIV and antiretroviral-associated MetS and lipodystrophy had increased EAT thickness and carotid intima–media thickness as compared with HIV-infected patients without MetS and lipodystrophy [91].

## 10. EAT in Psoriasis

Psoriasis is an immune-mediated skin disease associated with increased proinflammatory cytokines and the actions of T-helper-17 and T-helper-1 cells. Psoriasis is associated with systemic comorbidities including MetS and ASCVD. There is thought to be a crosstalk between skin inflammation and adipose tissue inflammation [92]. Increased EAT was first described as a marker of cardiovascular disease risk in psoriasis patients in a 2013 cross-sectional and observational study looking at 65 patients with psoriasis and 50 controls [93]. In a 2014 case-control study looking at 31 patients with psoriasis and 32 control subjects, EAT was found to be significantly higher in patients with psoriasis, and this was independent of pre-existing MetS [94]. In a study of 38 patients with psoriasis and 38 controls, EAT, as measured by CT, was higher in patients with psoriasis and EAT was independently associated with the presence of coronary calcium in all subjects [95]. Another study of 115 adults with psoriasis and 60 matched controls found that EAT as measured by transthoracic echocardiography was higher in the psoriasis group and also found that high-sensitivity CRP was higher in the psoriasis group [96]. A study of 100 patients with severe psoriasis and without ASCVD and 202 controls underwent CT to measure EAT, abdominal VAT, and coronary artery calcification found that psoriasis was associated with subclinical atherosclerosis via coronary artery calcification and with EAT independently of abdominal VAT [97]. A 2016 systematic review and meta-analysis including the five aforementioned trials concluded that psoriasis was associated with increased VAT [98].

## 11. Racial and Ethnic Differences in EAT

The difference in EAT thickness between subjects with and without MetS varies with ethnicity [50,99,100,101,102,103]. It is unclear whether this variability depends on racial differences in visceral adipose tissue amount, as reported in some ethnic groups, or on differences concerning the extent of visceral fat increase needed to trigger the MetS [99,100,101,102,103]. A 2012 meta-analysis of nine studies on the relationship between EAT and MetS found that the difference in EAT thickness between subjects with or without the MetS was more evident in Caucasian subjects, followed by Hispanic, Turkish, and Asian subjects [50].

In a study of 150 patients admitted to a clinical decision unit in Michigan for chest pain, EAT, as measured by echocardiography, was significantly greater in non-Hispanic White Caucasians compared with non-Hispanic Black African-Americans [99]. This was true even when adjusted for age, sex, BMI, and waist circumference [99]. Similar results were found in a study in Miami where EAT was measured by echocardiography [100] and another study in South Carolina where EAT was measured by CT even when adjusted for cardiovascular risk factors [101]. A larger study of 1199 middle-aged men (24.2% White, 7.0% Black, 23.6% Japanese-Americans, 22.0% Japanese, 23.2% Koreans) found that EAT volumes as measured by CT were highest among Japanese-Americans and lowest among Blacks [102]. Furthermore, associations of EAT with BMI and VAT differed by racial/ethnic groups. A 2017 Australian study of 150 subjects found that EAT was significantly higher in South Asians and Southeast or East Asians when compared with Whites [103]. South Asians were also found to have a higher aggregative plaque volume in the left anterior descending artery compared with Whites [103].

## 12. Conclusions with Future Perspectives

EAT is a measurable, modifiable potential therapeutic target that is correlated with VAT. Measurement of EAT by transthoracic echocardiography is a cost-effective, easily accessible, accurate and reproducible method of measuring EAT. Increased EAT is a risk factor for MetS and ASCVD. EAT is associated with T2DM, insulin resistance and liver steatosis. EAT is elevated in HIV patients on antiretroviral therapy and in psoriasis patients, two groups with increased risk of MetS. Normal EAT varies by race and ethnic group.

EAT measurement serves as a powerful potential diagnostic tool in assessing cardiovascular risk. In fact, measurement of VAT via EAT may one day serve as a useful diagnostic tool for risk stratification of MetS and its associated conditions including sleep apnea, atrial fibrillation, stroke, dementia, various cancers; and osteoporosis [26]. Furthermore, modification of EAT thickness via weight loss [104] and pharmacologic therapy has therapeutic implications for cardiovascular disease, T2DM, and MetS. We recommend the use of transthoracic echocardiography to measure EAT in patients with high-risk of ASCVD and MetS. We also recommend taking into account race and ethnicity when measuring EAT.

## Figures and Tables

**Figure 1 ijms-20-05989-f001:**
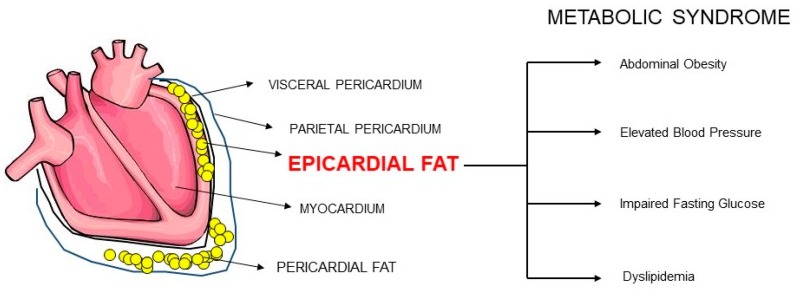
Epicardial fat is associated with metabolic syndrome.

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
