# Peer review of "Epicardial Adipose Tissue: Clinical Biomarker of Cardio-Metabolic Risk"

_ijms, 2019, doi:10.3390/ijms20235989_

Round 1

Reviewer 1 Report

Comments to Authors:

Fricke and Iacobellis, have conducted a scoping review study to examine the role of EAT in various facets of MetS including type 2 diabetes mellitus (T2DM), insulin resistance and liver steatosis. Firstly, the study was well designed and employed robust references list. Secondly, the aims are clear, and the conclusions are supported by the scoping review. Finally, EAT measurement serves as a powerful potential diagnostic tool in assessing cardiovascular.

However, some issues should be considered:

Measurement of EAT EAT as an endocrine organ EAT and new therapeutic target

Lastly, please include a cartoon (figure) that plausible shown role of EAT on MetS, T2DM, insulin resistance and liver steatosis.

Author Response

Thank you for your comments. The following sentences describing the properties of epicardial fat as endocrine organ was added in the introductory section ( page 2)

“EAT synthesizes, produces and secretes bio-active molecules which are then transported into the adjacent myocardium through vasocrine and/or paracrine pathways. Given the lack of anatomical barriers separating the two tissues, the EAT secretome goes directly into myocardium and coronary lumen.   Based on this evidence, EAT can be correctly considered an endocrine organ with local effects on the heart.”

Reviewer 2 Report

In this review Villasante Fricke and Iacobellis discuss the role of epicardial adipose tissue in some metabolic condiction related to insulin resistance, as well as the interestingly correlation of EAT with HIV therapy and psoriasis.

Could authors report the possible limitations of the transthoracic echocardiographic ultrasound in EAT evaluation? Particularly in patients with metabolic syndrome - overweight/obese (poor acoustic window for ultrasound) -  in which this exam should be recommended based on authors’ conclusions

Other comments:

1)      Please remove (:) in the title of each paragraphs.

2)      Line 100: duplication of “this”.

Author Response

Thank you for your comments. The following paragraph describing the limitations of echocardiographic epicardial fat measurement was added at the end of section “EAT as marker of visceral adiposity” (page 3).

"The ultrasound assessment of the EAT thickness has several intuitive advantages, but also some limitations. It provides a linear measurement rather than a volumetric one. The ultrasound is operator-dependent which could influence the reproducibility. Furthermore, EAT located in the atrioventricular groove or other locations cannot be reached with the ultrasound. Severely obese patients with MetS may have a poor acoustic window not allowing an optimal visualization of the EAT thickness. Multidetector CT and Cardiac MRI imaging can certainly provide a more accurate and volumetric measurement of EAT. EAT attenuation as measured by CT has also recently emerged as marker of adipose tissue inflammation and a potential predictor of CAD. However, these procedures are more invasive and costly, and may not be readily available for a rapid and practical VAT assessment in the MetS patients."

The colons were removed from the end of section titles. The duplicate "this" was removed.    

Reviewer 3 Report

This paper is a review on the role of epicardial adipose tissue (EAT) on cardio-metabolic risk. The relationship between EAT and conditions such as metabolitan syndrome, inflammatory status, diabetes, NAFLD and CV risk are clearly developed in the paragraphs of the text.
The review is complete and also expresses the authors' contribution to the development of knowledge on this topic. This also emerges from the numerous self-citations of the authors present among the references.
The paper will certainly be of interest to readers.

This reviewer raises only a few minor comments:
1- At the end of the paragraph "Adipose Inflammation in Metabolic Syndrome", I would add a sentence to point out that the opioid system also seems to play a role in insulin resistance in subjects with visceral obesity (ref. Cozzolino D. et al. The involvement of the opioid system in human obesity: A study in normal weight relatives of obese people Journal of Clinical Endocrinology and Metabolism Volume 81, Issue 2, 1996, Pages 713-718).
2- The manuscript would be enriched by one or two figures that illustrate the relationships described in the text between EAT and the cardio-metabolic risk.

Author Response

Thank you for your comments. We added the following sentence to the suggested paragraph on page 2:

Additionally, there is some evidence to suggest that opioids can trigger insulin resistance is people with a familial predisposition to obesity the beta-endorphin system [25].

Please see new table and figure.